# ACCOUNTING FOR UNOBSERVED CONFOUNDING IN DOMAIN GENERALIZATION

## ABSTRACT

The ability to extrapolate, or generalize, from observed to new related environments is central to any form of reliable machine learning, yet most methods fail when moving beyond $i.i.d$ data. In some cases, the reason lies in a misappreciation of the causal structure that governs the data, and in particular as a consequence of the influence of unobserved confounders that drive changes in observed distributions and distort correlations. In this paper, we argue for defining generalization with respect to a broader class of distribution shifts (defined as arising from interventions in the underlying causal model), including changes in observed, unobserved and target variable distributions. We propose a new robust learning principle that may be paired with any gradient-based learning algorithm. This learning principle has explicit generalization guarantees, and relates robustness with certain invariances in the causal model, clarifying why, in some cases, test performance lags training performance. We demonstrate the empirical performance of our approach on healthcare data from different modalities, including image and speech data.

## 1  INTRODUCTION

Prediction algorithms use data, necessarily sampled under specific conditions, to learn correlations that extrapolate to new or related data. If successful, the performance gap between these two domains is small, and we say that algorithms *generalize* beyond their training data. Doing so is difficult however, some form of uncertainty about the distribution of new data is unavoidable. The set of potential distributional changes that we may encounter is mostly unknown and in many cases may be large and varied. Some examples include covariate shifts (Bickel et al., 2009), interventions in the underlying causal system (Pearl, 2009), varying levels of noise (Fuller, 2009) and confounding (Pearl, 1998). All of these feature in modern applications, and while learning systems are increasingly deployed in practice, generalization of predictions and their reliability in a broad sense remains an open question.

A common approach to formalize learning with uncertain data is, instead of optimizing for correlations in a *fixed* distribution, to do so simultaneously for a *range* of different distributions in an uncertainty set $\mathcal{P}$ (Ben-Tal et al., 2009).

$$\underset{f}{\text{minimize}} \sup_{P \in \mathcal{P}} \mathbb{E}_{(x,y) \sim P}[\mathcal{L}(f(x), y)] \qquad (1)$$

for some measure of error $\mathcal{L}$ of the function $f$ that relates input and output examples $(x, y) \sim P$. Choosing different sets $\mathcal{P}$ leads to estimators with different properties. It includes as special cases, for instance, many approaches in domain adaptation, covariate shift, robust statistics and optimization (Kuhn et al., 2019; Bickel et al., 2009; Duchi et al., 2016; 2019; Sinha et al., 2017; Wozabal, 2012; Abadeh et al., 2015; Duchi & Namkoong, 2018). Robust solutions to problem (1) are said to generalize if potential shifted, test distributions are contained in $\mathcal{P}$, but also larger sets $\mathcal{P}$ result in *conservative* solutions (i.e. with sub-optimal performance) on data sampled from distribution away from worst-case scenarios, in general.

One formulation of causality is in fact also a version of this problem, for $\mathcal{P}$ defined as any distribution arising from *arbitrary* interventions on observed covariates $x$ leading to shifts in their distribution $P_x$ (see e.g. sections 3.2 and 3.3 in (Meinshausen, 2018)). The invariance to changes in covariate distributions of causal solutions is powerful for generalization, but implicitly assumes that all

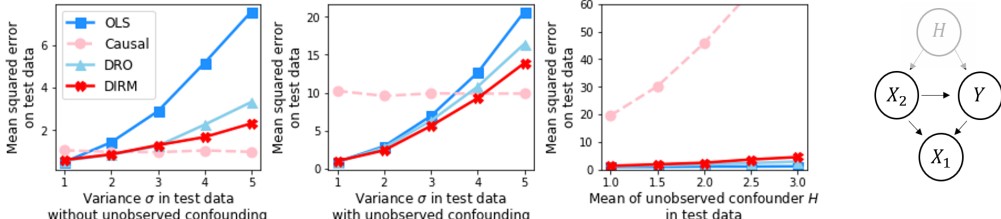

Figure 1: **The challenges of generalization**. In the presence of unobserved confounders, there is an inherent trade-off in performance – causal and correlation-based solutions are both optimal in different regimes, depending on the shift from which new data is generated. The proposed approach, DIRM, is a relaxation of the causal solution that naturally interpolates between the causal solution and Ordinary Least Squares (OLS), and is described in Section 3. The data generating mechanism, methods, and a discussion of the results are given in the paragraphs above our contributions.

covariates or other drivers of the outcome subject to change at test time are observed. Often shifts occur elsewhere, for example in the distribution of unobserved confounders, in which case also conditional distributions $P_{y|x}$ may shift. Perhaps surprisingly, in the presence of unobserved confounders, the goals of achieving robustness and learning a causal model can be *different* (and similar behaviour also occurs with varying measurement noise). There is in general an inherent trade-off in generalization performance. In the presence of unobserved confounders, causal and correlation-based solutions are both optimal in different regimes, depending on the shift in the underlying generating mechanism from which new data is generated.

Consider a simple example, illustrated in Figure 1, to show this explicitly. We assume access to observations of variables $(X_1, X_2, Y)$ in two training datasets, each dataset sampled with differing variances ($\sigma^2 = 1$ and $\sigma^2 = 2$) from the following structural model $\mathbb{F}$,

$$X_2 := -H + E_{X_2}, \quad Y := X_2 + 3H + E_Y, \quad X_1 := Y + X_2 + E_{X_1},$$

$E_{X_1}, E_{X_2} \sim \mathcal{N}(0, \sigma^2)$, $E_Y \sim \mathcal{N}(0, 1)$ are exogenous variables. In a first scenario (leftmost panel) we consider all data (training and testing) to be generated *without* unobserved confounders, $H := 0$; and, in a second scenario (remaining panels) all data *with* unobserved confounders, $H := E_H \sim \mathcal{N}(0, 1)$. Each panel of Figure 1 shows performance on *new* data obtained after manipulating the underlying data generating system; the magnitude and type of intervention appears in the horizontal axis. We consider the following learning paradigms: Ordinary Least Squares (OLS) learns the linear mapping that minimizes average training risk, Domain Robust Optimization (DRO) minimizes the maximum training risk among the two available datasets, and the causal solution, assumed known, has fixed coefficients $(0, 1)$ for $(X_1, X_2)$. Two important observations motivate this paper.

First, observe that Ordinary Least Squares (OLS) and Domain Robust Optimization (DRO) absorb *spurious* correlations (due to $H$, and the fact that $X_1$ is caused by $Y$) with unstable performance under shifts in $p(X_1, X_2)$ but as a consequence good performance under shifts in $p(H)$. Causal solutions, by contrast, are robust to shifts in $p(X_1, X_2)$, even on new data with large shifts, but underperform substantially under changes in the distribution of unobserved confounders $p(H)$. Second, the presence of unobserved confounding *hurts* generalization performance in general with higher errors for all methods, e.g. contrast the middle and leftmost panel. To the best of our knowledge, the influence of unobserved confounders has been minimally explored in the context of generalization of learning algorithms, even though, as Figure 1 shows, in this context different shifts in distribution may have important consequences for predictive performance.

**Our Contributions.** In this paper we provide a new choice of $\mathcal{P}$ and learning problem (1) that we show to be justified by certain statistical *invariances* across training and testing data, to be expected in the presence of unobserved confounders. This leads us to define a new differentiable, regularized objective for representation learning. Our proposal defines $\mathcal{P}$ as an *affine* combination of available training data distributions, and we show that solutions to this problem are robust to more general shifts in distribution than previously considered, spanning robustness to shifts in observed, unobserved, and target variables, depending on the properties of the available training data distributions. This approach has benefits for performance out-of-sample but also for tasks involving variable selection, where important features are consistently replicated across experiments with our objective.

## 2 INVARIANCES IN THE PRESENCE OF UNOBSERVED CONFOUNDERS

This section formally introduces the problem of out-of-distribution generalization. We describe in greater detail the reasons that popular learning principles, such as Empirical Risk Minimization (ERM), underperform in general, and define certain invariances to recover solutions that generalize.

We take the perspective that all potential distributions that may be observed over a system of variables arise from a causal model $\mathcal{M} = (\mathbb{F}, \mathbb{V}, \mathbb{U})$, characterized by endogenous variables, $\mathbb{V} \in \mathcal{V}$, representing all variables determined by the system, either observed or not; exogenous variables, $\mathbb{U} \in \mathcal{U}$, in contrast imposed upon the model, and a sequence of structural equations $\mathbb{F} : \mathcal{U} \to \mathcal{V}$, describing how endogenous variables can be (deterministically) obtained from the exogenous variables (Pearl, 2009). An example is given in Figure 1, $\mathbb{V} = (X_1, X_2, H, Y)$ are endogenous and $\mathbb{U} = (E_{X_1}, E_{X_2}, E_H, E_Y)$ are exogenous variables.

Unseen data is generated from such a system $\mathcal{M}$ after manipulating the distribution of exogenous variables $\mathbb{U}$, which propagates across the system shifting the joint distribution of all variables $\mathbb{V}$, whether observed or unobserved, but keeping the causal mechanisms $\mathbb{F}$ unchanged. Representative examples include changes in data collection conditions, such as due to different measurement devices, or new data sources, such as patients in different hospitals or countries, among many others.

Our goal is to learn a representation $Z = \phi(X)$ acting on a set observed variables $X \subset \mathbb{V}$ with the ability to extrapolate to new unseen data, and doing so acknowledging that all relevant variables in $\mathbb{V}$ are likely not observed. Unobserved confounders (for the task at hand, say predicting $Y \in \mathbb{V}$) simultaneously cause $X$ and $Y$, confounding or biasing the causal association between $X$ and $Y$ giving rise to spurious correlations that do not reproduce in general (Pearl, 1998; 2009). We present a brief argument below highlighting the systematic bias due to unobserved confounders in ERM.

### 2.1 THE BIASES OF UNOBSERVED CONFOUNDING

Consider the following structural equation for observed variables $(X, Y)$,

$$Y := f \circ \phi(X) + E \tag{2}$$

where $f := f(\cdot; \beta_0)$ is a predictor acting on a representation $Z := \phi(X)$ and $E$ stands for potential sources of mispecification and unexplained sources of variability. For a given sample of data $(x, y)$ and $z = \phi(x)$, the optimal prediction rule $\hat{\beta}$ is often taken to minimize squared residuals, with $\hat{\beta}$ the solution to the normal equations: $\nabla_\beta f(z; \hat{\beta}) y = \nabla_\beta f(z; \hat{\beta}) f(z; \hat{\beta})$, where $\nabla_\beta f(z; \hat{\beta})$ denotes the column vector of gradients of $f$ with respect to parameters $\beta$ evaluated at $\hat{\beta}$. Consider the Taylor expansion of $f(z; \beta_0)$ around an estimate $\hat{\beta}$ sufficiently close to $\beta_0$, $f(z; \beta_0) \approx f(z; \hat{\beta}) + \nabla_\beta f(z; \hat{\beta})^T (\beta_0 - \hat{\beta})$. Using this approximation in our first order optimality condition we find,

$$\nabla_\beta f(z; \hat{\beta}) \nabla_\beta f(z; \hat{\beta})^T (\beta_0 - \hat{\beta}) + v = \nabla_\beta f(z; \hat{\beta}) \epsilon \tag{3}$$

where $v$ is a scaled disturbance term that includes the rest of the linear approximation of $f$ and is small asymptotically; $\epsilon := y - f(z; \hat{\beta})$ is the residual. $\hat{\beta}$ is consistent for the true $\beta_0$ if and only if $\nabla_\beta f(z; \hat{\beta}) \epsilon \to 0$ in probability. This assumption is satisfied if $E$ (all sources of variation in $Y$ not captured by $X$) are independent of $X$ (i.e. exogenous) or in other words if all common causes or confounders to both $X$ and $Y$ have been observed. Conventional regression may assign significant associations to variables that are neither directly nor indirectly related to the outcome, and in this case, we have no performance guarantees on new data with changes in the distribution of these variables. Omitted variables are a common source of unobserved confounding but we note in Appendix B that similar biases also arise from other prevalent model mispecifications, such as measurement error (Carroll et al., 2006).

### 2.2 INVARIANCES WITH MULTIPLE ENVIRONMENTS

The underlying structural mechanism $\mathbb{F}$, that also relates unobserved with observed variables, even if unknown, is stable irrespective of manipulations in exogenous variables that may give rise to heterogeneous data sources. Under certain conditions, statistical footprints emerge from this structural invariance across different data sources, properties testable from data that have been exploited recently, for example (Peters et al., 2016; Ghassami et al., 2017; Rothenhäusler et al., 2019).

We assume that such a heterogeneous data scenario applies, input and output pairs $(X, Y)$ are observed across heterogeneous data sources or environments $e$, defined as a probability distribution $P_e$ over an observation space $\mathcal{X} \times \mathcal{Y}$ that arises, just like new unseen data, from manipulations in the distribution of exogenous variables in an underlying model $\mathcal{M}$.

For the remainder of this section, consider restricting ourselves to data sources emerging from manipulations in exogenous $E_X$, appearing in the structural equations of $X$ only in an underlying additive noise model (see Appendix C.1 for the precise statement of assumptions and more context). It may be shown by considering the distributions of error terms $Y - f \circ \phi(X)$ and its correlation with any function of $X$, that the inner product $\nabla_\beta f(z; \beta_0)\epsilon$, even if *non-zero* due to unobserved confounding, converges to a fixed unknown value *equal* across training environments (see Appendix C.1 for the derivation). With a similar decomposition to the one given in equation (3), in the population case, it holds that up to disturbance terms,

$$\left( \mathop{\mathbb{E}}_{(x,y)\sim P_i} \nabla_\beta f(z; \beta^\star)\nabla_\beta f(z; \beta^\star)^T - \mathop{\mathbb{E}}_{(x,y)\sim P_j} \nabla_\beta f(z; \beta^\star)\nabla_\beta f(z; \beta^\star) \right)^T (\beta_0 - \beta^\star)$$
$$= \left( \mathop{\mathbb{E}}_{(x,y)\sim P_i} \nabla_\beta f(z; \beta^\star)\epsilon - \mathop{\mathbb{E}}_{(x,y)\sim P_j} \nabla_\beta f(z; \beta^\star)\epsilon \right) = 0 \qquad (4)$$

where $\beta^\star$ is a solution to,

$$\mathop{\mathbb{E}}_{(x,y)\sim P_i} \nabla_\beta f(z; \beta)(y - f(z; \beta)) - \mathop{\mathbb{E}}_{(x,y)\sim P_j} \nabla_\beta f(z; \beta)(y - f(z; \beta)) = 0. \qquad (5)$$

and is *consistent* for the causal parameters $\beta_0$ if unique. $i, j \in \mathcal{E}$ are the indices of any two observed environments in an index set $\mathcal{E}$. This invariance across environments must hold for causal parameters (under certain conditions) *even* in the presence of unobserved confounders.

A few remarks are necessary concerning this relationship and its extrapolation properties.

- The first is based on the observation that, up to a constant, each inner product in (5) is the gradient of the squared error with respect to $\beta$. This reveals that the optimal predictor, in the presence of unobserved confounding, is not one that produces minimum loss but one that produces a *non-zero* loss gradient *equal* across environments. Seeking minimum error solutions, even in the population case, produces estimators with *necessarily* unstable correlations because the variability due to unobserved confounders is not explainable from observed data. Forcing gradients to be zero then *forces* models to utilize artifacts of the specific data collection process that are not related to the input-output relationship; and, for this reason, will not in general perform outside training data.

- From (5) we may pose a sequence of moment conditions for each pair of available environments. We may then seek solutions $\beta$ that make all of them small simultaneously. Solutions are unique if the set of moments is sufficient to identify $\beta^\star$ exactly (and given our model assumptions may be interpreted as causal and robust to certain interventions). We revisit our introductory example to show in Appendix A that, in contrast to ERM and Invariant Risk Minimization (IRM) (a related approach proposed in (Arjovsky et al., 2019) we discuss in more detail in later sections), this procedure does recover the underlying causal model correctly in the presence of unobserved confounding.

- In practice however, only a set of solutions may be identified with no performance guarantees for any individual solutions, and no guarantees if assumptions fail to hold. Moreover, even if accessible, causal solutions, robust to certain distribution shifts, may not always be desirable under more general shifts (recall for instance the experiments in the rightmost panel of Figure 1).

## 3 GENERALIZATION FROM A ROBUST OPTIMIZATION PERSPECTIVE

While certain invariances may hold in the presence of unobserved confounding, we have in general no guarantees on performance under more general manipulations. In this section we motivate a relaxation of the ideas presented above by considering bounds on the worst-case performance on data arising from distribution shifts informed by the structure of available environments.

Optimizing for the worst case loss in a set of domains has been shown to ensure accurate prediction on any convex mixture of training environments (Ben-Tal et al., 2009). The space of convex mixtures,

however, is restrictive. Systems of variables are in general high-dimensional and new manipulations likely occur at a new vertex not represented as a linear combination of training environments. By *extrapolation* we desire performance guarantees outside this convex hull. The extension we consider optimizes instead over an *affine* combination of training losses, similarly to (Krueger et al., 2020).

Let $\Delta_\eta := \{\{\alpha_e\}_{e \in \mathcal{E}} : \alpha_e \geq -\eta, \sum_{e \in \mathcal{E}} \alpha_e = 1\}$ be a collection of scalars and consider the set of distributions defined by $\mathcal{P} := \{\sum_{e \in \mathcal{E}} \alpha_e P_e : \{\alpha_e\} \in \Delta_\eta\}$, all affine combinations of distributions defined by the available environments. $\eta \in \mathbb{R}$ defines the strength of the extrapolation, $\eta = 0$ corresponds to a convex hull of distributions but above that value the space of distributions is richer, going beyond what has been observed: affine combinations amplify the strength of manipulations that generated the observed training environments. The following theorem presents an upperbound to the robust problem (1) with affine combinations of errors.

**Theorem 1** *Let $\{P_e\}_{e \in \mathcal{E}}$, be a set of available environments. Further let the parameter space of $\beta$ be open and bounded, such that the expected loss function $\mathcal{L}$ as a function of $\beta$ belongs to a Sobolev space. Then, the following inequality holds,*

$$\sup_{\{\alpha_e\} \in \Delta_\eta} \sum_{e \in \mathcal{E}} \alpha_e \mathop{\mathbb{E}}_{(x,y) \sim P_e} \mathcal{L}\left(f \circ \phi(x), y\right) \leq \mathop{\mathbb{E}}_{(x,y) \sim P_e, e \sim \mathcal{E}} \mathcal{L}\left(f \circ \phi(x), y\right)$$

$$+ (1 + n\eta) \cdot C \cdot \left\| \sup_{e \in \mathcal{E}} \mathop{\mathbb{E}}_{(x,y) \sim P_e} \nabla_\beta \mathcal{L}\left(f \circ \phi(x), y\right) - \mathop{\mathbb{E}}_{(x,y) \sim P_e, e \sim \mathcal{E}} \nabla_\beta \mathcal{L}\left(f \circ \phi(x), y\right) \right\|_{L_2}$$

*where $\| \cdot \|_{L_2}$ denotes the $L_2$-norm, $C$ depends on the domain of $\beta$, $n := |\mathcal{E}|$ is the number of available environments and $e \sim \mathcal{E}$ loosely denotes sampling indeces with equal probability from $\mathcal{E}$.*

The proof is given in Appendix C.2. This bound illustrates the trade-off between invariances (of the type explored under unobserved confounding, that appear in the second term of the RHS of the inequality above) and prediction in-sample (the first term). A combination of them upper-bounds a robust optimization problem over affine combinations of training environments, and depending how much we weight each objective (prediction versus invariance) we can expect solutions to be more or less robust. Specifically, for $\eta = -1/n$ the objective reduces to empirical risk minimization, but otherwise the upperbound increasingly weights differences in loss derivatives (violations of the invariances of section 2.2), and in the limit ($\eta \to \infty$) can be interpreted to be robust at least to *any* affine combination of training losses.

Note that the requirement that $\mathbb{F}$ be fixed, or interventions occur on observed variables, is not necessary for generalization guarantees. As long as new data distributions can be represented as affine combinations of training distributions, we can expected performance to be as least as good as that observed for the robust problem in Theorem 1.

### 3.1 PROPOSED OBJECTIVE

Our proposal is to guide the optimization of $\phi$ and $\beta$ towards solutions that minimize the upperbound in Theorem 1, satisfying approximately the moment conditions (5) and simultaneously optimizing for minimum average error. Using Lagrange multipliers we define the general objective,

$$\underset{\beta, \phi}{\text{minimize}} \mathop{\mathbb{E}}_{(x,y) \sim P_e, e \sim \mathcal{E}} \mathcal{L}\left(f \circ \phi(x), y\right) + \lambda \cdot \underset{e \sim \mathcal{E}}{\text{Var}} \left( \| \mathop{\mathbb{E}}_{(x,y) \sim P_e} \nabla_\beta \mathcal{L}\left(f \circ \phi(x), y\right) \|_{L_2} \right) \quad (6)$$

where $\lambda \geq 0$. We call this problem Derivative Invariant Risk Minimization (DIRM). This objective shares similarities with the objective proposed in (Krueger et al., 2020). The authors considered enforcing equality in environment-specific losses, rather than derivatives, as regularization, which can also be related to a robust optimization problem over an affine combination of errors. We have seen in section 2.2 however that equality in losses is not expected to hold in the presence of unobserved confounders (e.g. due to changes in the irreducible error across environments, which may occur also after interventions on target variables).

The $L_2$ norm in the regularizer is an integral over the domain of values of $\beta$ and is in general intractable. We approximate this objective in practice with norms on functional evaluations at each step of the optimization. Theorem 1 is used as a guide for optimization, we encourage the values of the regularizer to be zero and thus only indirectly the regularizer function itself. We give more details in Appendix D.1.1.

### 3.2 ROBUSTNESS IN TERMS OF INTERVENTIONS

As is apparent in Theorem 1, performance guarantees on data from a new environment depend on the relationship of new distributions with those observed during training.

Let $f \circ \phi_{\lambda \to \infty}$ minimize $\mathcal{L}$ among all functions that satisfy all pairs of moment conditions defined in (5); that is, a solution to our proposed objective in (6) with $\lambda \to \infty$. At optimality, it holds that gradients evaluated at this solution are equal across environments. As a consequence of Theorem 1, the loss evaluated at this solution with respect to *any* affine combination of environments is bounded by the average loss computed in-sample (denoted $L$, say),

$$\sum_{e \in \mathcal{E}} \alpha_e \mathop{\mathbb{E}}_{(x,y) \sim P_e} \mathcal{L}(f \circ \phi(x), y) \leq L, \qquad \text{for any set of } \alpha_e \in \Delta_\eta \tag{7}$$

From the perspective of interventions in the underlying causal mechanism, this can be seen as a form of data-driven predictive stability across a range of distributions whose perturbations occur in the same direction as those observed during training. As a minimal example for intuition, consider a system of three variables $(X, Y, H)$, with differing interventions on $X$ only. Assume we have access to data sampled under two environments defined by joint distributions $p_1(X, Y, H) := p(Y, H|X)p_1(X)$ and $p_2(X, Y, H) := p(Y, H|X)p_2(X)$. Errors are bounded with respect to any (valid) distribution of the form $p(Y, H|X)(\alpha_1 p_1(X) + \alpha_2 p_2(X))$. For instance, a particular shift in the mean or variance of $X$ observed during training can be extrapolated in the extreme to *any* shift in the mean or variance of new data. With this reasoning, if environment-specific distributions arise from differing interventions on all observed covariates $X$, then solutions are robust to arbitrary shifts in these variables. If, in addition, solutions are unique we may interpret them as causal, irrespective of the presence or not of unobserved confounders.

The generalization properties, however, go further. Interventions on unobserved variables and also target variables are accommodated for in $f \circ \phi_{\lambda \to \infty}$, if observed through shifted distributions in different available environments.

Using our simple example in Figure 1 to verify this intuition empirically, we consider 3 scenarios corresponding to interventions on exogenous variables of $X, H$ and $Y$. In each, training data from two environments is generated with means in the distribution of the concerned variables set to a value of 0 and 1 respectively, everything else being equal ($\sigma^2 := 1, H := E_H \sim \mathcal{N}(0, 1)$). Performance is evaluated on out-of-sample data generated by increasing the shift in the variable being studied up to a mean of 5. In all cases, we see in Figure 2 that performance is stable to increasing perturbations in the system as long as the heterogeneity in the data allows us to capture the direction of the unseen shift.

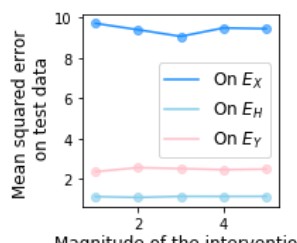

Figure 2: Stability to general shifts.

### 3.3 STABILITY OF CERTAIN OPTIMAL SOLUTIONS

A special case may also be considered when the underlying system of variables and the available environments allow for optimal solutions $f \circ \phi_{\lambda \to \infty}$ and $f \circ \phi_{\lambda=0}$ to coincide. In this case, the learned representation $\phi(x)$ results in a predictor $f$ optimal on average *and* simultaneously with equal gradient in each environment, thus,

$$\left|\left| \mathop{\mathbb{E}}_{(x,y) \sim P_e} \nabla_\beta \mathcal{L}(f \circ \phi(x), y) \right|\right|_{L_2} = 0, \qquad \text{for all } e \in \mathcal{E} \tag{8}$$

For this representation $\phi$, it follows that optimal solutions $f$ learned on any new dataset sampled from an affine combination of training distributions coincides with this special solution. This gives us a sense of reproducibility of learning. In other words, if a specific feature is significant for predictions on the whole range of $\lambda$ with the available data then it will likely be significant on new (related) data.

The above special case where all solutions in our hyperparameter range agree has important parallels with IRM (Arjovsky et al., 2019). The authors proposed a learning objective enforcing representations of data with minimum error on average and across environments, such that at optimum $\mathbb{E}_{P_i} Y | \phi^\star(X) = \mathbb{E}_{P_j} Y | \phi^\star(X)$ for any pair $(i, j) \in \mathcal{E}$. With unobserved confounding, both learning paradigms agree but, with unobserved confounding, minimum error solutions of IRM by design

converge to spurious associations (see the discussion after equation (5)) and are not guaranteed to generalize to more general environments. For example, in the presence of additive unobserved confounding $H$, irrespective of $\phi$, we may have $\mathbb{E}_{P_i} Y | \phi^\star(X) = \phi^\star(X) + \mathbb{E}_{P_i} H \neq \phi^\star(X) + \mathbb{E}_{P_j} H = \mathbb{E}_{P_j} Y | \phi^\star(X)$ if the means of $H$ differ. The sought invariance then does not hold.

## 4    RELATED WORK

**Causality**. There has been a growing interest in interpreting shifts in distribution to fundamentally arise from interventions in the causal mechanisms of data. Peters et al. (Peters et al., 2016) exploited this link for causal inference: causal relationships by definition being invariant to the observational regime. Invariant solutions, as a result of this connection, may be interpreted also as robust to certain interventions (Meinshausen, 2018), and recent work has explored learning invariances in various problem settings (Arjovsky et al., 2019; Rothenhäusler et al., 2019; Krueger et al., 2020; Gimenez & Zou, 2020). Among those, we note the invariance proposed in (Rothenhäusler et al., 2019), the authors seek to recover causal solutions with unobserved confounding. Generalization properties of these solutions were rarely studied, with one exception being Anchor regression (Rothenhäusler et al., 2018). The authors proposed to interpolate between empirical risk minimization and causal solutions with explicit robustness to certain interventions in a linear model. The present work may be interpreted as a non-linear formulation of this principle with a more general study of generalization.

**Domain generalization** represent one direction of out-of-sample generalization by explicitly learning representations projecting out superficial environment-specific information. Recent work on domain generalization has included the use data augmentation (Volpi et al., 2018; Shankar et al., 2018), meta-learning to simulate domain shift (Li et al., 2018) and adversarially learning representations that are environment invariant (Ganin et al., 2016), even though explicitly aligning representations has important caveats when label distributions differ, articulated for instance in (Arjovsky et al., 2019).

**Distributionally robust optimization** explicitly solves a worst-case optimization problem (1). A popular approach is to define $\mathcal{P}$ as a ball around the empirical distribution $\hat{P}$, for example using $f$-divergences or Wasserstein balls of a defined radius (Kuhn et al., 2019; Duchi et al., 2016; 2019; Sinha et al., 2017; Wozabal, 2012; Abadeh et al., 2015; Duchi & Namkoong, 2018). These are general and multiple environments are not required, but this also means that sets are defined agnostic to the geometry of plausible shifted distributions, and may therefore lead to solutions, when tractable, that are overly conservative or do not satisfy generalization requirements (Duchi et al., 2019).

## 5    EXPERIMENTS

Data linkages, electronic health records, and bio-repositories, are increasingly being collected to inform medical practice. As a result, also prediction models derived from healthcare data are being put forward as potentially revolutionizing decision-making in hospitals. Recent studies (Cabitza et al., 2017; Venugopalan et al., 2019), however, suggest that their performance may reflect not only their ability to identify disease-specific features, but also their ability to exploit spurious correlations due to unobserved confounding (such as varying data collection practices): a major challenge for the reliability of decision support systems. In this section, we explore this pattern conducting a wide analysis of domain generalization on image, speech and tabular data from the medical domain.

We consider the following baseline algorithms for performance comparisons.

- Empirical Risk Minimization (**ERM**) that optimizes for minimum loss agnostic of data source.
- Domain Robust Optimization (**DRO**) (Sagawa et al., 2019) that optimizes for minimum loss across the worst convex mixture of training environments.
- Domain Adversarial ENural Netowrks (**DANN**) (Ganin et al., 2016) that use domain adversarial training to facilitate transfer by augmenting the neural network architecture with an additional domain classifier to enforce the distribution of $\phi(X)$ to be the same across training environments.
- Invariant Risk Minimization (**IRM**) (Arjovsky et al., 2019) that regularizes ERM ensuring representations $\phi(X)$ be optimal in every observed environment.
- Risk Extrapolation (**REx**) (Krueger et al., 2020) that regularizes for equality in environment losses instead of considering their derivatives.

|  | Pneumonia Prediction | | Parkinson Prediction | | Survival Prediction | |
|---|---|---|---|---|---|---|
|  | Training | Testing | Training | Testing | Training | Testing |
| ERM | 91.4 ($\pm$ .7) | 52.4 ($\pm$ 1) | 95.7 ($\pm$ .5) | 62.9 ($\pm$ 1) | 93.2 ($\pm$ .4) | 75.3 ($\pm$ .9) |
| DRO | 91.2 ($\pm$ .5) | 53.1 ($\pm$ .6) | 94.0 ($\pm$ .3) | 69.8 ($\pm$ 2) | 90.5 ($\pm$ .4) | 75.5 ($\pm$ .8) |
| DANN | 92.3 ($\pm$ 1) | 57.1 ($\pm$ 2) | 91.6 ($\pm$ 2) | 51.4 ($\pm$ 5) | 89.3 ($\pm$ .8) | 73.9 ($\pm$ .9) |
| IRM | 89.5 ($\pm$ 1) | 58.6 ($\pm$ 2) | 93.6 ($\pm$ 1) | 71.4 ($\pm$ 2) | 91.9 ($\pm$ .6) | 75.7 ($\pm$ .8) |
| REx | 87.7 ($\pm$ 1) | 57.9 ($\pm$ 2) | 92.0 ($\pm$ 1) | 72.4 ($\pm$ 2) | 91.3 ($\pm$ .5) | 75.0 ($\pm$ .9) |
| DIRM | 84.3 ($\pm$ 1) | 63.7 ($\pm$ 3) | 93.1 ($\pm$ 2) | 72.8 ($\pm$ 2) | 91.4 ($\pm$ .6) | 77.9 ($\pm$ 1) |

Table 1: Accuracy of predictions in percentages (%). Uncertainty intervals are standard deviations. All datasets are approximately balanced, 50% performance is as good as random guessing.

All trained models use the same convolutional or fully-connected architecture, where appropriate. Performance results are given in Table 1. Further experimental details and pseudo-code for DIRM can be found in Appendix D.

## 5.1 DIAGNOSIS OF PNEUMONIA WITH CHEST X-RAY DATA

In this section, we attempt to replicate the study in (Zech et al., 2018). The authors observed a tendency of image models towards exploiting spurious correlations for the diagnosis on pneumonia from patient Chest X-rays that do not reproduce outside of training data. We use publicly available data from the National Institutes of Health (NIH) (Wang et al., 2017) and the Guangzhou Women and Children's Medical Center (GMC) (Kermany et al., 2018). Differences in distribution are manifest, and can be seen for example in the top edge of mean pneumonia-diagnosed X-rays shown in Figure 3.

In this experiment, we exploit this (spurious) pathology correlation to demonstrate the need for solutions robust to changes in site-specific features. We construct two training sets, in each case 90% and 80% of pneumonia-diagnosed patients were drawn from the NIH dataset and the remaining 10% and 20% of the pneumonia-diagnosed patients were drawn from the GMC dataset; the reverse logic (10%/90% split) was followed for the test set.

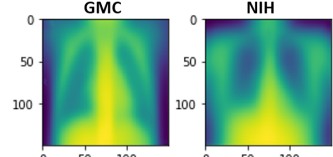

Figure 3: Average pneumonia X-ray.

Our results show that DIRM outperforms, suggesting that the proposed invariances guides solutions better towards robustness even to changes due to unobserved factors.

## 5.2 DIAGNOSIS OF PARKINSON'S DISEASE WITH VOICE RECORDINGS

Parkinson's disease is a progressive nervous system disorder that affects movement. Symptoms start gradually, sometimes starting with a barely noticeable tremor in a patient's voice. This section investigates the performance of predictive models for the detection of Parskinson's disease, trained on voice recordings of vowels, numbers and individual words and tested on vowel recordings of unseen patients.

We used the UCI Parkinson Speech Dataset with given training and testing splits (Sakar et al., 2013). Even though the distributions of features will differ in different types of recordings and patients, we would expect the underlying patterns in speech to reproduce across different samples. However, this is not the case for correlations learned with baseline training paradigms (Table 1). This suggests that spurious correlations due to the specific type of recording (e.g. different vowels or numbers), or even chance associations emphasized due to low sample sizes (120 examples), may be responsible for poor generalization performance. Our results show that correcting for spurious differences between recording types (DIRM, IRM, REx) can improve performance substantially.

## 5.3 SURVIVAL PREDICTION WITH ELECTRONIC HEALTH RECORDS

This section investigates whether predictive models transfer across data from different medical studies (the MAGGIC, 2012, studies), all containing patients that experienced heart failure. The problem is to predict survival within 3 years of experiencing heart failure from a total of 33 demographic variables. We introduce a twist however, explicitly introducing unobserved confounding by omitting certain

predictive variables. The objective is to test performance on new studies with *shifted* distributions, while knowing that these occur predominantly due to variability in unobserved variables.

Confounded data is constructed by omitting a patient's age from the data, found in a preliminary correlation analysis to be associated with the outcome as well as other significant predictors such as blood pressure and body mass index. This example is constructed to be able to control for how unobserved variables shift but note that we can expect similar phenomena in many other scenarios, where for instance a prediction model is taken to patients in a different hospital or country with fundamental shifts in the distribution of very relevant variables (e.g. socio-economic status, ethnicity, diet, etc.) even though this information is not reported in the data. Performance is tested on all studies of over 500 patients with balanced death rates, each having slightly different age distributions. (We give more details in Appendix D). We found DIRM, robust to changes in unobserved variables, to outperform all other methods.

**Influential variables that reproduce across datasets.** In the following, we tackle the problem of *reproducibility* of learned influential features across different experiments. Reproducing conclusions of influential features in different studies with potential shifts in the distribution is an important challenge, especially in healthcare where heterogeneity between patient populations is high. We showed in section 3.3 that in the event that the optimal predictor is invariant as a function $\lambda \in [0, \infty)$, optimal predictors estimated in *every* new dataset in the span of observed distributions, should be *stable*.

We consider here a form of diluted stability for feature selection. For a single layer network, we consider significant those covariates with estimated parameters bounded away from zero in all solutions in the range $\lambda \in [0, 1]$. Comparisons are made with ERM (conventional logistic regression), both methods trained separately on 100 random pairs of studies. Figure 4 shows how many features (in the top 10 of predictive features) from each model intersect across pairs of studies. In constrast to ERM, our objective recovers significant features much more consistently serves to demonstrate the improved reproducibility we can expect from DIRM.

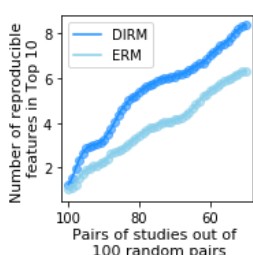

Figure 4: Reproducible features.

# 6 CONCLUSIONS

We have studied the problem of out-of-sample generalization from a new perspective, grounded in the underlying causal mechanism generating new data that may arise from shifts in observed, unobserved or target variables. Our proposal is a new objective that is provably robust to certain shifts in distribution, and is informed by certain statistical invariances in the presence of unobserved confounders. Our experiments show that we may expect better generalization performance and also better reproducibility of influential features in problems of variable selection.

A limitation of our approach is that robustness guarantees crucially depend on the (unobserved) properties of available data. Using the proposed approach, Derivative Invariant Risk Minimization for prediction generally does not guarantee protection against unsuspected events. More specifically, we can not expect robust prediction when the heterogeneity in test data sets is different from the restricted set of shift interventions that have been observed on the training data sets. For example, in Theorem 1, the supremum contains distributions that lie in the affine combination of training environments, as opposed to arbitrary distributions.

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

# Appendix

This Appendix contains 4 sections.

- Section A presents additional experiments designed to demonstrate the causal interpretation one may give to DIRM if all conditions for causality are satisfied (i.e. are available interventions on all observed variables). We also include a sensitivity analysis to show the impact on performance of changing the regularization parameter $\lambda$.

- Section B shows how measurement error may be interpreted as an instance of unosebrved confounding.

- Section C provides proofs for the statements made in the main body of this paper.

- Section D gives additional experimental details, including on the implementation of DIRM and on the datasets used.

## A  ADDITIONAL EXPERIMENTS

So far, we have considered predictive performance under different data distributions with selected hyper-parameter configurations of all algorithms to illustrate heterogeneous behaviour of algorithms trained with different learning principles in the presence of unobserved confounders. In this section we revisit our introductory example to investigate in more details learned prediction rules and any sensitivities of interest, especially to hyper-parameter configurations.

We will use the same data generating mechanism presented in the introductory example in Figure 1 of the main body of this paper. Recall that we assume access to observations of variables $(X_1, X_2, Y)$ in two training datasets, each dataset sampled with differing interventions on $(X_1, X_2)$ (in this case differing variances $\sigma^2 = 1$ and $\sigma^2 = 2$) from the following structural model,

$$X_2 := -H + E_{X_2}, \quad Y := X_2 + 3H + E_Y, \quad X_1 := Y + X_2 + E_{X_1} \quad H := E_H$$

where $E_{X_1}, E_{X_2} \sim \mathcal{N}(0, \sigma^2)$, $E_Y \sim \mathcal{N}(0, 1)$, $E_H \sim \mathcal{N}(0, 1)$ are exogenous variables. $H$ is an unobserved confounder, not observed during training but that influences the observed association between $X_2$ and $Y$.

### A.1  RECOVERY OF CAUSAL COEFFICIENTS

In this section, given the above two training datasets, we inspect the weights learned in a simple one layer feed-forward neural network to determine exactly whether unobserved confounding induces a given learning paradigm to exploit spurious correlations and to what extent.

By way of preface, we have mentioned that causal, in contrast with spurious, solutions to a prediction problem may be defined as the argument solving,

$$\underset{f}{\text{minimize}} \, \sup_{P \in \mathcal{P}} \, \mathbb{E}_{(x,y) \sim P}[\mathcal{L}(f(x), y)] \tag{9}$$

for $\mathcal{P}$ defined as any distribution arising from *arbitrary* interventions on observed covariates $x$ leading to shifts in their distribution $P_x$ (see sections 3.2 and 3.3 in (Meinshausen, 2018) for a detailed discussion of this result). This objective is a special case of the proposed optimization problem (6), specifically it is an affine combination (with $\lambda \to \infty$) of distributions with different shifts in $P_x$ in all observed variables $x$.

We demonstrate this fact empirically in Table 2. In principle, causal solutions are recoverable with the proposed approach because we do observe during training environments with shifts in $p(X_1, X_2)$, irrespective of the presence or not of unobserved confounders. We see that this holds approximately for the proposed objective with estimated coefficients $(0.01, 0.95)$ for $(X_1, X_2)$ close to the true causal coefficients $(0, 1)$. In contrast, ERM returns biased coefficients and so does IRM.

This empirical observation is important because it highlights the fact that enforcing minimum gradients on average (ERM) or simultaneously across environments (the regularization proposed by IRM) is not appropriate to recover causal coefficients in the presence of unobserved confounders.

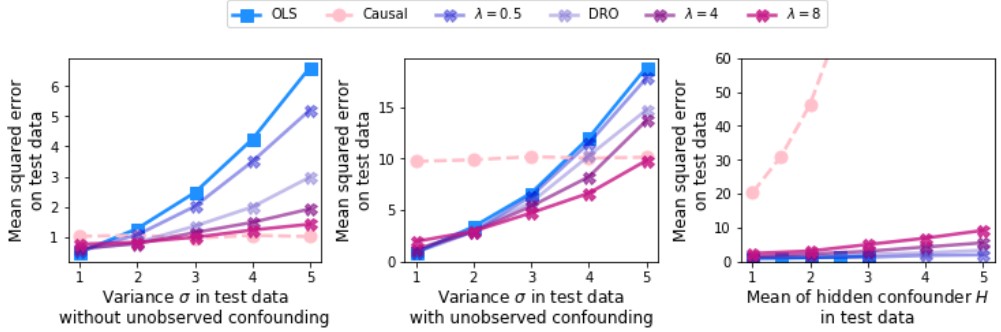

Figure 5: **Sensitivity of solutions to hyperparameter** $\lambda$. Ordinary Least Squares (OLS) and the causal solution, with coefficients $(0, 1)$ for $(X_1, X_2)$ are two extremes ($\lambda = 0$ and $\lambda \to \infty$ respectively) of the spectrum of solutions that can be attained with the proposed approach. Positive values of $\lambda$ interpolate in some sense between OLS and causal solutions in this case. Here DRO corresponds to DIRM with $\lambda = 2$, approximately.

If however, no unobserved confounders exist in the system being modelled ($H := 0$ in the data generating mechanism) our objective and IRM are equivalent in the limit, and estimated parameters coincide with the causal solution approximately. This experiment is given in Table 3.

|  | **Truth** | **ERM** | **IRM** ($\lambda \to \infty$) | **DIRM** ($\lambda \to \infty$) |
|---|---|---|---|---|
| Estimated parameters | [0, 1] | [0.91, -1.02] | [0.75, -0.76] | [0.01, 0.95] |

Table 2: Bias in estimation **with** unobserved confounders.

|  | **Truth** | **ERM** | **IRM** ($\lambda \to \infty$) | **DIRM** ($\lambda \to \infty$) |
|---|---|---|---|---|
| Estimated parameters | [0, 1] | [0.5, -0.6] | [0.01, 0.98] | [0.02, 0.96] |

Table 3: Bias in estimation **without** unobserved confounders.

## A.2 SENSITIVITY TO HYPER-PARAMETERS

The robustness guarantees of any particular solution depends on the extent of the extrapolation desired (as a function of $\lambda$). For larger values of this parameter we can expect solutions to be robust in a larger set of distributions, spanning empirical risk minimization for $\lambda = 0$, to convex combinations, to training environments to arbitrary affine combinations of training environments for increasing $\lambda$.

In this section, we analysed performance in test data with the exact same data generating mechanism as considered in the introduction of the main body of this paper as a function of $\lambda$. Figure 5 gives our performance results that empirically verifies that the proposed approach interpolates between empirical risk minimization and causality in this case. We can see that for $\lambda$ approaching zero solutions converge to ERM, for $\lambda = 2$ the solutions was equivalent to DRO, and for increasing $\lambda$ the solutions approximate the causal one in the limit.

## B OTHER EXAMPLES OF UNOBSERVED CONFOUNDING

**Measurement error.** The data generating processes described in the main body of this paper for instance, as well as most of machine learning, assume that all nuisance variability enters the causal mechanisms of the data; that is, observed data reflects *only* causal drivers. If this is not the case, for example because of independent measurement noise observed in data but that does not propagate to across causal children, regression is known to be inconsistent in general (Carroll et al., 2006) and its bias is analogous to a form of unobserved confounding.

Consider a simple model for illustration. Suppose $(X, Y)$ are observed subject to measurement noise, $X^\star = X + E_x$ and $Y^\star = Y + E_y$, which are not causally related to one another but rather $Y = \beta X + E$. Let $E_x = \beta_x H$ and $E_y = \beta_y H$ be the structure of measurement error independent of $X$ and $Y$. Then substituting our observed data $(X^\star, Y^\star)$ into the underlying $(X, Y)$ relationship the observed model is,

$$Y^\star = \beta X^\star + (\beta_y - \beta_x\beta)H + E, \quad X^\star = \beta_x H + X \tag{10}$$

A special case of regression with unobserved confounders $H$.

## C    TECHNICAL RESULTS

This section provides a more complete discussion of the assumptions and justification statements relating to causality in section 2.2, and the proof of Theorem 1.

### C.1    INVARIANCES IN THE PRESENCE OF UNOBSERVED CONFOUNDING

In section 2.2 we justified exploiting a certain invariance of causal coefficients in the inner product of functions of the data $X$ and residuals $E$, to occur even in the presence of unobserved confounders as long as interventions that define different environments do not involve unobserved confounders $H$.

Here we show this invariance to hold in the special case of an additive model. The general data generation mechanism is as follows. Data sources, or different environments, emerge from manipulations in exogenous $E_X$, related to $X$ only, in an underlying additive model $\mathbb{F}$ with also additive functions $f_1, f_2, f_3, f_4$,

$$Y := f_1(X) + f_2(H) + E_Y, \qquad X := f_3(X) + f_4(H) + E_X, \qquad H := E_H \tag{11}$$

Exogenous variables $(E_X, E_Y, E_H)$ may have arbitrary distributions but only $E_X$ or $E_Y$ vary across environments. Then it holds that,

$$X = (I - f_3)^{-1}f_4(H) + (I - f_3)^{-1}E_X$$
$$= (I - f_3)^{-1}f_4(E_H) + (I - f_3)^{-1}E_X$$

and that,

$$\nabla_\beta f_1(X)(Y - f_1(X)) = \left(\nabla_\beta f_1(I - f_3)^{-1}f_4(E_H) + \nabla_\beta f_1(I - f_3)^{-1}E_X\right) \cdot (f_2(E_H) + E_Y)$$

which is a product of functions involving $E_H$ in one term, $E_H$ and $E_Y$ in another term, $E_X$ and $E_H$ in another term, and $E_X$ and $E_Y$ in the last term. Since $(E_X, E_Y, E_H)$ are mutually independent taking expectations of product of functions involving $E_X$ and $E_H$, $E_X$ and $E_H$, and, $E_X$ and $E_Y$ equals 0 assuming $f_i(E_j) = 0$ for $i = 1, \ldots, 4$ and $j \in \{X, Y, H\}$.

So concluding, the expectation of the inner product $\nabla_\beta f_1(X)(Y - f_1(X))$ does not depend on $E_X$ nor $E_Y$ and is thus stable across environments that have changing distributions for $E_X$ or $E_Y$. Now note that other functions than $f_1$ may have this property as well, i.e. predictors that satisfy this invariance are not necessarily unique and will depend on the differences between available environments. If however, only one predictor exist that satisfies this invariance we may say that this predictor is causal. We summarize this claim in the following statement.

**Proposition 1** *Let $Y$ and $X$ be related by a non-linear additive model with unobserved confounding as in (11). Then,*

$$\mathbb{E}_{P_i}\nabla_\beta f_1(X)(Y - f_1(X)) = \mathbb{E}_{P_j}\nabla_\beta f_1(X)(Y - f_1(X)) \tag{12}$$

*under the assumption that distributions on $(X, Y)$ $P_i$ and $P_j$ are given by a data generating mechanism (11) subject to interventions on $E_X$ or $E_Y$ only. Moreover, a function $f$ satisfying the above equality, if unique is equal to $f_1$.*

## C.2 PROOF OF THEOREM 1

We restate the Theorem for convenience.

**Theorem 1** *Let $\{P_e\}_{e \in \mathcal{E}}$, be a set of available environments. Further let the parameter space of $\beta$ be open and bounded, such that the expected loss function $\mathcal{L}$ as a function of $\beta$ belongs to a Sobolev space. Then, the following inequality holds,*

$$\sup_{\alpha_e \in \Delta_\eta} \sum_{e \in \mathcal{E}} \alpha_e \mathop{\mathbb{E}}_{(x,y) \sim P_e} \mathcal{L}\left(f \circ \phi(x), y\right) \leq \mathop{\mathbb{E}}_{(x,y) \sim P_e, e \sim \mathcal{E}} \mathcal{L}\left(f \circ \phi(x), y\right)$$

$$+ (1 + n\eta) \cdot C \cdot \left|\left| \sup_{e \in \mathcal{E}} \mathop{\mathbb{E}}_{(x,y) \sim P_e} \nabla_\beta \mathcal{L}\left(f \circ \phi(x), y\right) - \mathop{\mathbb{E}}_{(x,y) \sim P_e, e \sim \mathcal{E}} \nabla_\beta \mathcal{L}\left(f \circ \phi(x), y\right) \right|\right|_{L_2}$$

*where $C$ depends on the domain of $\beta$, $n := |\mathcal{E}|$ is the number of available environments and $e \sim \mathcal{E}$ loosely denotes sampling indeces with equal probability from $\mathcal{E}$.*

*Proof.* Let $\Omega$ denote the parameter space of $\beta$. The following derivation shows the claim,

$$\sup_{\alpha_e \in \Delta_\eta} \sum_{e \in \mathcal{E}} \alpha_e \mathop{\mathbb{E}}_{(x,y) \sim P_e} \mathcal{L}\left(f \circ \phi(x), y\right)$$

$$= (1 + n\eta) \cdot \sup_{e \in \mathcal{E}} \mathop{\mathbb{E}}_{(x,y) \sim P_e} \mathcal{L}\left(f \circ \phi(x), y\right) - \eta \sum_{e \sim \mathcal{E}} \mathbb{E}_{P_e} \mathcal{L}\left(f \circ \phi(x), y\right)$$

$$= \mathop{\mathbb{E}}_{(x,y) \sim P_e, e \sim \mathcal{E}} \mathcal{L}\left(f \circ \phi(x), y\right) + (1 + n\eta) \cdot \sup_{e \in \mathcal{E}} \mathop{\mathbb{E}}_{(x,y) \sim P_e} \mathcal{L}\left(f \circ \phi(x), y\right)$$

$$- (\eta + 1/n) \sum_{e \sim \mathcal{E}} \mathop{\mathbb{E}}_{(x,y) \sim P_e} \mathcal{L}\left(f \circ \phi(x), y\right)$$

$$= \mathop{\mathbb{E}}_{(x,y) \sim P_e, e \sim \mathcal{E}} \mathcal{L}\left(f \circ \phi(x), y\right)$$

$$+ (1 + n\eta) \cdot \left( \sup_{e \in \mathcal{E}} \mathop{\mathbb{E}}_{(x,y) \sim P_e} \mathcal{L}\left(f \circ \phi(x), y\right) - \mathop{\mathbb{E}}_{(x,y) \sim P_e, e \sim \mathcal{E}} \mathcal{L}\left(f \circ \phi(x), y\right) \right)$$

$$\leq \mathop{\mathbb{E}}_{(x,y) \sim P_e, e \sim \mathcal{E}} \mathcal{L}\left(f \circ \phi(x), y\right)$$

$$+ (1 + n\eta) \cdot M \cdot \left|\left| \sup_{e \in \mathcal{E}} \mathop{\mathbb{E}}_{(x,y) \sim P_e} \mathcal{L}\left(f \circ \phi(x), y\right) - \mathop{\mathbb{E}}_{(x,y) \sim P_e, e \sim \mathcal{E}} \mathcal{L}\left(f \circ \phi(x), y\right) \right|\right|_{L_2}$$

where the inequality is given by the property that the evaluation functional is a bounded linear operator in certain Sobolev spaces $\mathcal{W}$, for example with $\Omega = \mathbb{R}^d$ and $L_2$ norm. In particular this means that $|f(\beta)| \leq M||f||_{L_2}$ for all $f \in \mathcal{W}$. It follows then also that the above is,

$$\leq \mathop{\mathbb{E}}_{(x,y) \sim P_e, e \sim \mathcal{E}} \mathcal{L}\left(f \circ \phi(x), y\right) +$$

$$(1 + n\eta) \cdot P \cdot M \cdot \left|\left| \sup_{e \in \mathcal{E}} \mathop{\mathbb{E}}_{(x,y) \sim P_e} \nabla_\beta \mathcal{L}\left(f \circ \phi(x), y\right) - \mathop{\mathbb{E}}_{(x,y) \sim P_e, e \sim \mathcal{E}} \nabla_\beta \mathcal{L}\left(f \circ \phi(x), y\right) \right|\right|_{L_2}$$

by Poincaré's inequality for Sobolev functions defined on an open, bounded parameter space, see e.g. (Leoni, 2017). The assumption we make here for this last inequality to hold is that the region where the difference in loss functions is near zero is large enough such that the integral of the gradient is also large enough to control the integral of the function. This holds however for functions defined on many "reasonable" parameter spaces (Lipschitz suffices).

## D EXPERIMENTAL DETAILS

This section gives implementation details of DIRM, additional experiments to test sensitivities relating to optimization choices, and a complete description of the data and experiments performed in the main body of this paper.

### D.1 IMPLEMENTATION DETAILS

The regularizer in DIRM's objective in equation (6) controls the regularity or variation of the prediction function and encourages to learn a representation $\phi$ that results in a prediction function with similar derivatives in all training domains. The $L_2$ norm integrates out the influence of $\beta$ in the regularizer and thus most of the optimization involves $\phi$, though $\beta$ still plays a role in the first term of the objective.

In all our experiments, $f : \mathbb{R}^h \to \mathbb{R}$ as well as $\phi : \mathbb{R}^d \to \mathbb{R}^h$ are implemented as fully connected neural networks ($\phi$ with optional hidden layers). $\beta$ can thus be interpreted as the weights and biases of $f$. The $L_2$ norm must be approximated in practice, which we do by evaluating the vector norm of the derivative of $f$ with respect to $\beta$ on a batch of training examples of each environment. The variance on the computed norms between environments is a proxy for the maximum deviation between environments with a smoother gradient vector field. Each step of the optimization then alternates between an update on $\phi$ and update on $f$, as detailed in the algorithm below.

DIRM is sensitive to initialization and to the choice of hyperparameters – specifically its optimization schedule. In our experiments, we found best performance by increasing the relative weight of the penalty term $\lambda$ after a fixed number of iterations (and similar implementations are used for IRM and REx that suffer from similar challenges). This we believe could be a significant limitation for its use in practice since this choice must be made a priori. We investigated the sensitivity of DIRM to this optimization schedule in Table 4 that shows test accuracy as a function of the iteration at which penalty term weight $\lambda$ is increased.

|  | 2 epochs | 4 epochs | 6 epochs | 8 epochs | 10 epochs | 12 epochs |
|---|---|---|---|---|---|---|
| IRM | 56.4 ($\pm$ 6) | 58.1 ($\pm$ 3) | 59.2 ($\pm$ 3) | 59.1 ($\pm$ 2) | 58.1 ($\pm$ 3) | 57.8 ($\pm$ 1) |
| REx | 55.3 ($\pm$ 8) | 57.9 ($\pm$ 4) | 60.6 ($\pm$ 3) | 60.5 ($\pm$ 2) | 57.7 ($\pm$ 3) | 54.7 ($\pm$ 2) |
| DIRM | 54.1 ($\pm$ 7) | 61.7 ($\pm$ 4) | 63.8 ($\pm$ 3) | 63.5 ($\pm$ 3) | 62.6 ($\pm$ 2) | 58.2 ($\pm$ 1) |

Table 4: Test set performance (accuracy in %) on X-ray data as a function of the number of epochs used to increase penalty $\lambda$.

Choosing this number accurately is important for generalization performance. If $\lambda$ is increased too early, different initialization values (and the complex loss landscape) lead to different solutions with unreliable performance and a large variance. This happens for all methods. An initial number of iterations minimizing loss in-sample improves estimates for all methods which then converge to solutions that exhibit lower variance.

---

**Algorithm 1** DIRM

---

**Input:** datasets $\mathcal{D}_1, \ldots, \mathcal{D}_E$ in $E$ different environments, parameter $\lambda$, batch size $K$

**Initialize:** neural network model parameters $\phi, \beta$

**while** convergence criteria not satisfied **do**

    **for** $e = 1, \ldots, E$ **do**

        Estimate loss $\mathcal{L}_e(\phi, \beta)$ empirically using a batch of $K$ examples from $\mathcal{D}_e$.

        Estimate derivatives $\nabla_\beta \mathcal{L}_e(\phi, \beta)$ empirically using a batch of $K$ examples from $\mathcal{D}_e$.

    **end for**

    Update $\beta$ by stochastic gradient descent with,

$$\nabla_\beta \left( \frac{1}{E} \sum_{e=1}^{E} \mathcal{L}_e(\phi, \beta) \right)$$

    Update $\phi$ by stochastic gradient descent with,

$$\nabla_\phi \left( \frac{1}{E} \sum_{e=1}^{E} \mathcal{L}_e(\phi, \beta) + \lambda \cdot \mathrm{Var}(||\nabla_\beta \mathcal{L}_1(\phi, \beta)||_2^2, \ldots, ||\nabla_\beta \mathcal{L}_E(\phi, \beta)||_2^2) \right)$$

**end while**

---

### D.1.1 APPROXIMATION OF THE $L_2$ NORM IN PRACTICE

The bound given in Theorem 1 quantifies the discrepancy between function derivatives using the $L_2$ norm, defined as an intregal over possible parameter values $\beta$. For neural networks, computation of the $L_2$ norm is largely intractable and specifically, for networks of depth greater or equal to 4, it is an NP-hard problem (see Proposition 1 in (Triki et al., 2017)). Some approximation is thus unavoidable. One option is to recognise the $L_2$ norm as an expectation over functional evaluations, $||f||_{L_2} = \mathbb{E}_{x \sim \mathcal{U}(\Theta)} \left[ ||f(x)||_2^2 \right]^{1/2}$ for a continuous function $f$ taking values $x$ sampled uniformly from its domain $\Theta$. Empirical means are tractable yet they induce a much higher computational burden as these must be computed in every step of the optimization. Our approach is to take this approximation to its limit, making a single function evaluation at each step of the optimization using the current estimate $\beta$, as written in Algorithm 1.

This approximation loosens the connection between the bound given in Theorem 1 and the proposed algorithm. It remains justified however from a conceptual perspective as the objective of controlling an $L_2$ type of norm is to encourage the regularizer function towards 0, and thus the values of the regularizer (which we do explicitly). For empirical comparisons with the empirical mean approach, we implemented empirical means using all combinations of parameter values chosen from a grid of 5 parameter values around the current estimate $\beta$, $\{0.25\beta, 0.5\beta, \beta, 2\beta, 4\beta\}$. Table 5 shows similar performance across the real data experiments considered in the main body of this paper. A single evaluation is in practice enough to monitor invariance of representations to environment-specific loss derivatives.

|  | **Pneumonia Prediction** | **Parkinson Prediction** | **Survival Prediction** |
|---|---|---|---|
| DIRM-means | 63.5 ($\pm$ 3) | 73.0 ($\pm$ 1.5) | 78.0 ($\pm$ .9) |
| DIRM-single | 63.7 ($\pm$ 3) | 72.8 ($\pm$ 2) | 77.9 ($\pm$ 1) |

Table 5: Test set performance (accuracy in %) on real datasets for two different regularizer approximations to the $L_2$ norm.

## D.2 DATA DETAILS

### D.2.1 X-RAY DATA

We create training environments with different proportions of $X$-rays from our two hospital sources to induce a correlation between the hospital (and its specific data collection procedure) and the pneumonia label. The objective is to encourage learning principles to exploit a spurious correlation, data collection mechanisms should not be related to the probability of being diagnosed with pneumonia. The reason for creating two training data sets with slightly different spurious correlation patterns is to nevertheless leave a statistical footprint in the distributions to disentangle stable (likely causal) and unstable (likely spurious). In each of the training and testing datasets we ensured positive and negative labels remained balanced. The training datasets contained 2002 samples each and the testing dataset contained 1144 samples.

All learning paradigms trained a convolutional neural network, 2 layers deep, with each layer consisting of a convolution (kernel size of 3 and a stride of 1). All predictions were made off of the deepest layer of the network. The number of input channels was 64, doubled for each subsequent layer, and dropout was applied after each layer. We optimize the binary cross-entropy loss using Adam (learning rate 0.001) without further regularization on parameters and use Xavier initialization. While learning with IRM and the proposed approach, the respective penalty $\lambda = 1$ is added to the loss after 5 epochs of learning with $\lambda = 0$. Experiments are run for a maximum of 20 epochs with early stopping based on validation performance. All results are averaged over 10 trials with different random splits of the data, and the reported uncertainty intervals are standard deviations of these 10 performance results.

### D.2.2 PARKINSON'S DISEASE SPEECH DATA

The data includes a total of 26 features recorded on each sample of speech and set training and testing splits which we use in our experiments. For each patient 26 different voice samples including sustained vowels, numbers, words and short sentences where recorded, which we considered to be different but related data sources. We created three training environments by concatenating features from three number recordings, concatenating features from three word recording and concatenating features from three sentences; for a total of 120 samples in each of the three training environment. The available testing split contained 168 recordings of vowels, which we expect to differ from training environments because these are different patients and do not contain numbers or words. Positive and negative samples were balanced in both training and testing environments.

On this data, for all learning paradigms we train a multi-layer perceptron with two hidden layers of size 64 with tanh activations and dropout $(p = 0.5)$ after each layer. As in the image experiments, we optimize the binary cross-entropy loss using Adam (learning rate 0.001), $L_2$ regularization on parameters and use Xavier initialization. While learning with IRM and the proposed approach, the respective penalty is added to the loss $\lambda = 1$ after 200 epochs of learning with $\lambda = 0$ to ensure stable optimization. Experiments are run for a maximum of 1000 epochs with early stopping based on the validation performance. All results are averaged over 10 trials with different random seeds of our algorithm. This is to give a sense of algorithm stability rather than performance stability.

### D.2.3 MAGGIC ELECTRONIC HEALTH RECORDS

MAGGIC stands for Meta-Analysis Global Group in Chronic Heart Failure. The MAGGIC meta-analysis includes individual data on $39,372$ patients with heart failure (both reduced and preserved left-ventricular ejection fraction), from 30 cohort studies, six of which were clinical trials. $40.2\%$ of patients died during a median follow-up of 2.5 years. For our purposes, we removed patients that were censored or lost to follow-up to ensure well-defined outcomes after 3 years after being discharged from their respective hospitals. A total of 33 variables describe each patient including demographic variables: age, gender, race, etc; biomarkers: blood pressure, haemoglobin levels, smoking status, ejection fraction, etc; and details of their medical history: diabetes, stroke, angina, etc.

To curate our training and testing datasets, we proceeded as follows. On all patients follow-up over 3 years, we estimated feature influence of survival status after three years. A number of variables were significantly associated with survival out of which we chose Age, also found correlated with BMI and a number of medical history features, as a confounder for the effect of these variables on survival. We

used three criteria to select studies: having more than 500 patients enrolled and balanced death rates (circa $50\%$). 5 studies fitted these constraints: 'DIAMO', 'ECHOS', 'HOLA', 'Richa', 'Tribo'. Each was chosen in turn as a target environment with models trained on the other 4 training environments.

**Feature reproducibility experiments**. A natural objective for the consistency of health care and such that we may reproduce the experiments and their results in different scenarios is to find relevant features that are not specific to an individual medical study, but can also be found (replicated) on other studies with different patients. Heterogeneous patients and studies, along with different national guidelines and standards of care make this challenging. In our experiments we made comparisons of reproducibility in parameter estimates for models trained using Empirical Risk Minimization (ERM) and DIRM. We chose networks with a single layer with logistic activation and focused on the estimation of parameter to understand the variability in training among different data sources. Naturally, feature importance measured by parameter magnitudes makes sense only after normalization of the covariates to the same (empirical) variance (equal to 1) in each study separately. After this preprocessing step, for both ERM and the proposed approach we trained separate networks on 100 random pairs of studies (each pair concatenated for ERM) and returned the top 10 significant features (by the magnitude of parameters). Over all sets of significant parameters we then identified how many intersected across a fixed number of the 100 runs.

The same architecture and hyperparameters as in Parkinson's disease speech data experiments was used for MAGGIC data except that we increase the maximum training epochs to 5000.

