# OpenReview forum: "Accounting for Unobserved Confounding in Domain Generalization"
_ICLR.cc/2021/Conference — Reject_

### Official Review · AnonReviewer3 · 2020-10-27
**A new objective for out-of-sample generalization via causal invariances.**

**Rating:** 7
**Confidence:** 2

**Review:**

The authors propose an optimization objective for out-of-sample generalization that aims to exploit statistical structure that emerges from underlying causal mechanisms and is hence transferable across domains. Effectiveness of the approach is demonstrated on several real-world datasets.
The problem is thoroughly motivated and relevant literature reviewed. Experiments support the authors claims, although real-world examples seems somewhat contrived.
The paper lacks an investigation of where and how the proposed methodology fails. It would have been helpful to provide more intuition around the formal explanations in section 2 and 3.


## Detailed Comments
- "new or related data" -> what does "related data" mean?
- "Doing so is difficult however, some form of uncertainty about the distribution of new data is unavoidable." -> Please check grammar
- (eq1): why are we taking the *supremum* of the expected loss over distributions?
- "to problem (9)" -> I think you mean (1). Prolly the eqs (1) and (9) have the same latex equation label.
- acronym DRO is not defined in text (only in figure caption)

---

> ### Author Response · Authors · 2020-11-18
> **Thank you for your comments, and especially for highlighting the missing discussion on the limitations of DIRM**
>
> Thank you for your comments and for pointing out typos which have been corrected in the revised manuscript. Sorry for the slow reply. Please find below our answer to your questions and details on the corresponding changes we have in the revised manuscript.
>
> - The supremum can be understood as a maximum since our set of distributions is finite, suprema are generally used for these statements because they are well-defined even on infinite sets as long as they are upper-bounded.
> - We have included a paragraph in the conclusion that highlights what performance guarantees can and cannot be expected in practice. Specifically, we must have observed a certain shift between two training environments to be able to extrapolate to arbitrary shifts in that direction. We cannot for instance guarantee prediction performance in “black swan events” due to a shift that has not been observed before.

---

### Official Review · AnonReviewer1 · 2020-10-28
**tackle the unobserved confounding to address cross-domain generalization**

**Rating:** 5
**Confidence:** 4

**Review:**

summary:
This paper proposes a new domain generalization (DG) method, which enjoys certain statistical invariance property in the presence of unobserved confounders. The method is motivated by causal understanding of the underlying generating distribution and it assumes that the distribution generating the unseen data is obtained by manipulating the distribution of exogenous variables in the causal model.


pros:
- this work views the distribution family of training data as generated from manipulating a causal model $M$, which is novel in DG methods and of pracitical significance.

- the idea is well motivated (section 2) and clearly presented (section 3)

- related works are properly mentioned and discussed


cons:
- the considered distribution family seems a bit restrictive as it requires manipulation only on exogeneous variables in additive structural equations.

- the algorithm is not available in both the main text and appendix.

- the proposed DG method is not empirically studied on widely used benchmark data sets.


detailed comments:
- It seems the method can only handle confounder of two variables with an additive structureal euqation relating them. If that is the case, it would be better to elaborate corresponding requirements in numbered assumptions.

- The derivation of the last inequality in the proof of theorem 1 is not very straightforward. It would be better to elaborate its derivation in the appendix.

- Since the analysis throughout the main text assumes a given $\phi(x)$ but one has to learn both $f$ and $\phi$ in practice, it would be better to give the algorithm used in experimental section in the main text.

- Experiments on popular DG data sets (e.g., VLCS) are missing in this work. It would be better to show the performance of the proposed method on benchmark data set. If possible, it would  be great to also consider comparing with REx [1].

minor:
- typo under Eq. (5), *indeces*

[1] David Krueger, Ethan Caballero, Joern-Henrik Jacobsen, Amy Zhang, Jonathan Binas, Remi Le Priol, and Aaron Courville. Out

---

> ### Author Response · Authors · 2020-11-13
> **Update to the response and details on the changes made in the revised manuscript.**
>
> Thank you very much for your comments. Please find below a slightly updated review and changes made in the revised manuscript based on your suggestions.
>
> **On assumptions for causality.**
> - The comment of the distribution family of environments relates to causal inference and the moment conditions proposed in section 2.2. The assumptions there parallel those of Invariant Causal Prediction [1] (which also form the basis for IRM) but importantly relaxes that of causal sufficiency, i.e. invariances hold even if not all causes of Y are observed, including confounders (influencing two or more variables). This is a significant change with respect to IRM, the latter explicitly shown to be inadequate in the presence of unobserved confounders (and shown empirically in experiments given in Appendix A.1). Our derivation however does require additive noises and interventions on all observed variables, which we agree is still restrictive.
> - Part of our contribution is to acknowledge this fact (mentioned in the last bullet point of section 2.2) and propose an objective that leverages these invariances to achieve more general robustness guarantees. Specifically, with the objective we propose, even if not all conditions for causality hold, invariances are still useful to regularize prediction algorithms, in the sense that solutions can be expected to be robust to certain shifts in the data distribution.
>
> *Changes to the manuscript*: We have included pseudo-code in Appendix A.3 and described therein more details on our implementation. REx has been implemented with a similar optimization schedule as IRM and DIRM, and our results in the updated Table 1 show performance close to IRM, and competitive but below DIRM for all experiments. Thank you for pointing out the typo which will be corrected in the revised manuscript. Further validation of DIRM on domain generalization benchmark datasets is an important task, but we chose to defer it to future work because these datasets do not highlight bias due to unobserved confounding explicitly which is the characteristic of data we study in this work.
>
> [1] Peters, Jonas, Peter Buhlmann, and Nicolai Meinshausen. "Causal inference by using invariant prediction: identification and confidence intervals”. Series B Statistical methodology. (2016).

---

### Official Review · AnonReviewer2 · 2020-10-29
**Excellent paper with a few clarity issues**

**Rating:** 9
**Confidence:** 4

**Review:**

--- Update after discussion ---

After looking at the concerns raised by the other reviewers and the author responses, I have the following comments:

1. I think the that authors have more than adequately addressed the concerns raised by reviewer 4. With that said, I agree with reviewer 4 in that the link between theorem 1 and the proposed objective and its approximation is not made sufficiently clear in the text.

2. I whole-heartedly agree with reviewer 1 that a clear and concise list of the assumptions made would improve the paper.

Overall, however, I think the paper should still be accepted and will keep my original rating.

--- Original review ---

Overall, I found this to be an excellent paper. The topic of generalizing to new environments is clearly important and the authors do a good job motivating this problem. I found the paper well written and clear, with many intuitive examples. I found the method compelling and the theory appears correct. The experiments were well designed and convincing. Insofar as I have concerns with the paper, they relate to clearly communicating the specific setting considered by the authors and contrasting their work with others (details below).

--- Comments ---

1. One piece that was unclear to me was why if was necessary to assume that $\mathbb{F}$ remains fixed. Shifts in $\mathbb{F}$ are certainly possible in real applications (e.g., consider a change in a treatment policy relating observed lab measurements to observed treatments at a particular hospital). Is this a constraint on the method? That is, if the observed environments contain shifts in $\mathbb{F}$, would the proposed method fail to produce a model that is robust to those shifts?

2. More generally, I didn't think the assumptions were clearly communicated. I think paragraph 2 of A.3.1 should probably be moved into the main paper.

3. I found the example in the intro a bit unclear. I would try to communicate earlier what you are hoping to show with the example. Additionally, at this point in the paper, it is not really clear what a "causal solution" is or how it differs from the proposed solution.

4. I thought the remarks following Equation (5) were very helpful and would recommend adding a similar high-level discussion after Theorem 1. Something like: The first term on the RHS is the expected loss and the second term is zero if the constraints discussed above are satisfied, thus by minimizing the constrained expected loss, we are minimizing an upper bound on the LHS.

5. By the time I got to Equation (6) I found myself wondering why *this* robust objective is better than all the others. This was then addressed, in part, by the last parts of 3.1 and 3.3, but I would consider including a more explicit contrast between these various objectives earlier in the paper. I think something like Section 2 of Kreuger et al. (2020) would help contextualize your contribution a bit better.

6. Links to Equation (9) should be swapped for Equation (1) (e.g., page 1, par 2).

7. I would recommend swapping $\alpha$ for another symbol since $\alpha_e$ is also used.

---

> ### Author Response · Authors · 2020-11-18
> **Thank you for your comments and many suggestions**
>
> Thank you very much for your comments and many suggestions. Apologies for the slow reply. Please find below our answers to some of your questions, and the corresponding changes we have made to the revised manuscript.
>
> **Clarifications.**
> - The requirement that $\mathbb F$ be fixed is not necessary for generalization guarantees. As long as new data distributions can be represented as affine combinations of training distributions, we can expect performance to be as least as good as that observed for the robust objective. It is, however, necessary that $\mathbb F$ be fixed to interpret the DIRM solution causally.
>
> *Changes to the manuscript*:
> - The observation above has been emphasized before section 3.1 in the revised manuscript.
> - We have also revised our discussion of the introductory example, more clearly communicating the objective of Figure 1 which is two-fold: first, highlighting that generalization performance depends on the nature of the shift occurring in test data and that different methods behave very differently for different shifts. And second, highlighting that DIRM can be understood as interpolating between causal and correlation-based solutions in this example.
> - We have clarified the meaning of each term in Theorem 1.
> - Some of the equations were mislabelled which has been corrected, and $\alpha$ has been changed for $\eta$ for clarity.
> - In the revised manuscript, we now refer more explicitly to the Appendix to expand on the assumptions needed for causality in section 2.2. We chose not to expand on this further in the main body of the paper since these assumptions are not necessary for the subsequent results in section 3.

---

### Official Review · AnonReviewer4 · 2020-10-30
**Accounting for Unobserved Confounding in Domain Generalization**

**Rating:** 3
**Confidence:** 5

**Review:**

Summary: This paper proposes a new regularizer that can be plugged in gradient-based learning algorithms, which aims at solving the problems induced by unobserved confounders. And the authors provide the upper bound for one specific kind of distributionally robust optimization problem, whose uncertainty set is defined as the affine combinations of training distributions. And based on this the algorithm is proposed to deal with the problem of unobserved confounders. Experiments on three medical datasets validate the effectiveness of the method.

Strengths:
1.	The authors provide the upper bound of a group-DRO-like problem whose uncertainty set is the affine combination of training environments.
2.	The authors provide the moment conditions for each pair of environments under linear settings with unobserved confounders and show that the gradients should not be forced to be zero.
3.	Three medical experiments validate the effectiveness of the proposed method.


Weaknesses:
In spite of the strengths mentioned above, there are a few questions that are confusing.
1.	As for the simulated experiment: What is the purpose of the third figure in Figure 1? It shows that the perfect causal model performs bad under unobserved, while the other three methods performs almost the same. Further, the performance of the proposed DIRM and DRO is quite similar in this setting, which does not account for the effectiveness of the method. Besides, the result of IRM for this experiment is missed.
2.	As for the theoretical analysis:
a)	For Theorem 1, the right hand equation uses L_2 norm of a function of beta. I read the prove and I think this norm is defined as an integral which has nothing do with beta any more. Therefore, I wonder what does the regularizer proposed in equation(6) means since beta has already been integrated.
b)	For Theorem 1, the core assumption is ‘the expected loss function as a function of beta belongs to a Sobolev space’, which is confusing. Could you provide some explanations of this assumption or give some examples of it?
c)	Theorem 1 provides an upper bound for one specific kind of DRO problem whose uncertainty set is formulated as an affine combination of training distributions. However, in this article, the authors do not state what is the definition of the invariance here and why solve such DRO problem could achieve the invariance.
3.	As for the proposed objective function:
a)	As mentioned above, the L_2 norm is taken over a function of beta, which I think is not the Euclidean norm of the vector. Beta has already been integrated and this regularizer has nothing do with beta. I wonder how to compute this when optimizing?
b)	I wonder how this objective function can be optimized efficiently? The first concern is mentioned above as the computation of L_2 norm. The second concern is how to optimize the variance which is non-convex and hard to optimize. Namkoong et al. [1] convert the optimization of a variance-regularized problem to a f-divergence DRO for better optimization, while in this paper the authors take the opposite way. I wonder is there any theoretical guarantee of the optimization of the objective function(6).
4.	As for the experiments:
a)	The experimental results on the last two datasets are not convincing enough to validate the effectiveness of the proposed method, since the performance is similar to IRM, which I wonder if it is caused by the problems mentioned above(in 3).

[1] Duchi, J. , & Namkoong, H. . (2016). Variance-based regularization with convex objectives.

---

> ### Comment · AnonReviewer2 · 2020-11-11
> **RE: the dependence of the regularizer on $\beta$ (weakness 2a)**
>
> I understood this as trying to find a representation $\phi$ such that the optimal predictor given that representation is domain invariant. That is, it is ok that the regularizer does not depend on $\beta$ since it is attempting to constrain $\phi$, though I could certainly be wrong in this understanding.

---

> ### Author Response · Authors · 2020-11-13
> **Updated response including changes made to the revised manuscript**
>
> Thank you very much for your comments. Please see below an updated response, highlighting the changes we have made in the revised manuscript to address your concerns and include your suggestions.
>
> **Purpose of the third panel in Figure 1.**
> - The third panel of Figure 1 illustrates that causal solutions, while robust to shifts in the distribution of observed variables, underperform with shifts in unobserved confounders since by definition causal solutions do not capture any of the observed correlation due to these variables. In the presence of unobserved confounders, Figure 1 shows that there is a general trade-off in performance: causal solutions outperform with large shifts on observed variables while correlation-based methods (OLS) outperform with moderate shifts in observed variables and with shifts in unobserved variables. In this context, one interpretation of DIRM is as an interpolation between causal and correlation-based solutions. DRO (using convex combinations of environments) also proposes such an interpolation and can be interpreted as a special case of DIRM for a certain hyperparameter $\alpha$ (this has been changed to $\eta$ in the revised manuscript). To show more contrast with DRO we have update Figure 3 in the Appendix to include DRO.
>
> *Changes in the manuscript*: The discussion of Figure 1 has been improved to emphasize two things: 1) Causal and correlation-based solutions are optimal under different perturbation regimes – no single approach outperforms, and the difference in performance depends on which variables are intervened on in test data. 2) DIRM is designed to interpolate between causal and correlation-based solutions, a fact that is also shown in Figure 3 of the Appendix which includes comparisons with DRO.
>
> **On the regularizer of the objective and implementation.**
> - The variance term controls the regularity of the prediction function and its purpose is to encourage representations $\phi$ that lead to predictors with similar derivatives in all training environments (as described by Reviewer 2). Most of the optimization thus involves $\phi$, though $\beta$ still plays a role in the first term of the objective. The regularizer in practice is approximated by the (squared) vector norm of the derivative, evaluated at the current estimate of $\beta$ on a batch of training examples of each environment before taking the variance between norms (chosen as a proxy for the maximum deviation between environments because of its smoother gradient vector field).
> - The reason for defining the expected loss function to live in a Sobolev space is to ensure its partial derivatives have well-defined L_2 norms but is not otherwise constraining.
> - As to the question related to invariances, the DRO problem in Theorem 1 does not immediately achieve invariances. This is our interpretation upon seeing the resemblance between the last terms on the RHS of Theorem 1 and the moment conditions discussed in section 2.2. By analogy to causality as a DRO problem (i.e. where the uncertainty set contains arbitrary interventions on observed variables), taking $\lambda\rightarrow\infty$ (which corresponds to uncertainty sets with increasing perturbation magnitude) Theorem 1 shows that causal predictors are intrinsically linked with predictors that achieve some form of invariance between environments. This is discussed in Section 3.2, where we also note that the DRO problem as defined in Theorem 1 is more general than simply a proxy for causality. If the available environments span interventions on unobserved or target variables, then it shows that some form of invariance will be satisfied to arbitrary shifts in those variables. We observe this to hold approximately in Figure 2, albeit with a toy experiment.
> - DIRM is sensitive to initialization and to the choice of hyperparameters – specifically its optimization schedule. In our experiments, we found best performance by increasing the penalty term weight $\lambda$ after a fixed number of iterations. Since this choice must be made a priori, this could be a significant limitation for its use in practice, even though reasonable guesses can be made with held-out test sets which lead to good performance. This fact, however, applies to also to most out-of-sample generalization algorithms with tuning parameters, including IRM, REx and others.
>
> *Changes to the manuscript*: This discussion, as well as pseudo-code for DIRM, has been included in Appendix A.3. To further investigate the sensitivity of DIRM (as well as IRM and REx) to the optimization schedule we included also an additional experiment that shows test accuracy as a function of the iteration at which penalty term weight is increased. This can also be found in section A.3.

---

### Decision · Program_Chairs · 2021-01-07
**Final Decision**

**Decision:**

Reject

**Comment:**

This paper deals with domain generalization with causal modeling. Specifically, it considers a broader class of distribution shifts, arising from the system intervention perspective, and proposes some robust learning principle to achieve domain generalization. The paper is well written and has some interesting ideas. However, as pointed by Reviewers #1 and #4, the exact problem setting should be made more explicit, the theory and algorithm should be more consistent, and some very relevant contributions in the literature should be discussed or compared with.